# Do Wider Neural Networks Really Help Adversarial Robustness?

**Boxi Wu** *
State Key Lab of CAD&CG
Zhejiang University
boxiwu@zju.edu.cn

**Jinghui Chen** *
Pennsylvania State University
State College, PA 16801
jzc5917@psu.edu

**Deng Cai**
State Key Lab of CAD&CG
Zhejiang University
dengcai@cad.zju.edu.cn

**Xiaofei He**
State Key Lab of CAD&CG
Zhejiang University
xiaofeihe@cad.zju.edu.cn

**Quanquan Gu**
Dept. of Computer Science
UCLA
qgu@cs.ucla.edu

## Abstract

Adversarial training is a powerful type of defense against adversarial examples. Previous empirical results suggest that adversarial training requires wider networks for better performances. However, it remains elusive how does neural network width affect model robustness. In this paper, we carefully examine the relationship between network width and model robustness. Specifically, we show that the model robustness is closely related to the tradeoff between natural accuracy and perturbation stability, which is controlled by the robust regularization parameter $\lambda$. With the same $\lambda$, wider networks can achieve better natural accuracy but worse perturbation stability, leading to a potentially worse overall model robustness. To understand the origin of this phenomenon, we further relate the perturbation stability with the network's local Lipschitzness. By leveraging recent results on neural tangent kernels, we theoretically show that wider networks tend to have worse perturbation stability. Our analyses suggest that: 1) the common strategy of first fine-tuning $\lambda$ on small networks and then directly use it for wide model training could lead to deteriorated model robustness; 2) one needs to properly enlarge $\lambda$ to unleash the robustness potential of wider models fully. Finally, we propose a new Width Adjusted Regularization (WAR) method that adaptively enlarges $\lambda$ on wide models and significantly saves the tuning time.

## 1   Introduction

Researchers have found that Deep Neural Networks (DNNs) suffer badly from adversarial examples [59]. By perturbing the original inputs with an intentionally computed, undetectable noise, one can deceive DNNs and even arbitrarily modify their predictions on purpose. To defend against adversarial examples and further improve model robustness, various defense approaches have been proposed [48, 42, 18, 40, 67, 26, 58, 55]. Among them, adversarial training [23, 41] has been shown to be the most effective type of defenses [5]. Adversarial training can be seen as a form of data augmentation by first finding the adversarial examples and then training DNN models on those examples. Specifically, given a DNN classifier $f$ parameterized by $\boldsymbol{\theta}$, a general form of adversarial training with loss function

---

*Equal contribution.

35th Conference on Neural Information Processing Systems (NeurIPS 2021).

$\mathcal{L}$ can be defined as:

$$\underset{\boldsymbol{\theta}}{\operatorname{argmin}} \frac{1}{N} \sum_{i=1}^{N} \Big[ \underbrace{\mathcal{L}(\boldsymbol{\theta}; \mathbf{x}_i, y_i)}_{\text{natural risk}} + \lambda \cdot \underbrace{\max_{\widehat{\mathbf{x}}_i \in \mathbb{B}(\mathbf{x}_i, \epsilon)} \big[ \mathcal{L}(\boldsymbol{\theta}; \widehat{\mathbf{x}}_i, y_i) - \mathcal{L}(\boldsymbol{\theta}; \mathbf{x}_i, y_i) \big]}_{\text{robust regularization}} \Big], \qquad (1.1)$$

where $\{(\mathbf{x}_i, y_i)_{i=1}^n\}$ are training data, $\mathbb{B}(\mathbf{x}, \epsilon) = \{\widehat{\mathbf{x}} \mid \|\widehat{\mathbf{x}} - \mathbf{x}\|_p \leq \epsilon\}$ denotes the $\ell_p$ norm ball with radius $\epsilon$ centered at $\mathbf{x}$, and $p \geq 1$, and $\lambda > 0$ is the regularization parameter. Compared with standard empirical risk minimization, the extra robust regularization term encourages the data points within $\mathbb{B}(\mathbf{x}, \epsilon)$ to be classified as the same class, i.e., encourages the predictions to be stable. The regularization parameter $\lambda$ adjusts the strength of robust regularization. When $\lambda = 1$, it recovers the formulation in [41], and when $\lambda = 0.5$, it recovers the formulation in [23]. Furthermore, replacing the loss difference in robust regularization term with the KL-divergence based regularization recovers the formulation in [70].

One common belief in the practice of adversarial training is that, compared with the standard empirical risk minimization, adversarial training requires much wider neural networks to achieve better robustness. [41] provided an intuitive explanation: robust classification requires a much more complicated decision boundary, as it needs to handle the presence of possible adversarial examples. However, it remains elusive how the network width affects model robustness. To answer this question, we first examine whether the larger network width contributes to both the natural risk term and the robust regularization term in (1.1). Interestingly, when tracing the value changes in (1.1) during adversarial training, we observe that the value of the robust regulariza-

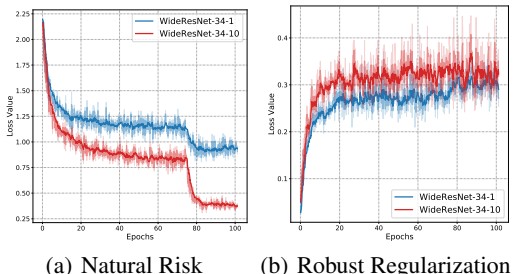

(a) Natural Risk     (b) Robust Regularization

Figure 1: Plots of both natural risk and robust regularization in (1.1). Two 34-layer WideResNet [69] are trained by TRADES [70] on CIFAR10 [37] with widen factor being 1 and 10.

tion part actually gets worse on wider models, suggesting that larger network width does not lead to better stability in predictions. In Figure 1, we show the loss value comparison of two different wide models trained by TRADES [70] with $\lambda = 6$ as suggested in the original paper. We can see that the wider model (i.e., WideResNet-34-10) achieves better natural risk but incurs a larger value on robust regularization. This motivates us to find out the cause of this phenomenon.

In this paper, we study the relationship between neural network width and model robustness for adversarially trained neural networks. Our contributions can be summarized as follows:

1. We show that the model robustness is closely related to both natural accuracy and perturbation stability, a new metric we proposed to characterize the strength of robust regularization. The balance between the two is controlled by the robust regularization parameter $\lambda$. With the same value of $\lambda$, the natural accuracy is improved on wider models while the perturbation stability often worsens, leading to a possible decrease in the overall model robustness. This suggests that proper tuning of $\lambda$ on wide models is necessary despite being extremely time-consuming, while directly using the fine-tuned $\lambda$ on small networks to train wider ones, as many people did in practice [41, 70], may lead to deteriorated model robustness.

2. Unlike previous understandings that there exists a trade-off between natural accuracy and robust accuracy, we show that the real trade-off should between natural accuracy and perturbation stability. And the robust accuracy is actually the consequence of this trade-off.

3. To understand the origin of the lower perturbation stability of wider networks, we further relate perturbation stability with the network's local Lipschitznesss. By leveraging recent results on neural tangent kernels [36, 3, 73, 8, 21], we show that with the same value of $\lambda$, larger network width naturally leads to worse perturbation stability, which explains our empirical findings.

4. Our analyses suggest that to unleash the potential of wider model architectures fully, one should mitigate the perturbation stability deterioration and enlarge robust regularization parameter $\lambda$ for training wider models. Empirical results verified the effectiveness of this strategy on benchmark datasets. In order to alleviate the heavy burden for tuning $\lambda$ on wide models, we develop the

Width Adjusted Regularization (WAR) method to transfer the knowledge we gain from fine-tuning smaller networks into the training of wider networks and significantly save the tuning time.

**Notation.** For a $d$-dimensional vector $\mathbf{x} = [x_1, ..., x_d]^\top$, we use $\|\mathbf{x}\|_p = (\sum_{i=1}^d |x_i|^p)^{1/p}$ with $p \geq 1$ to denote its $\ell_p$ norm. $\mathbb{1}(\cdot)$ represents the indicator function and $\forall$ represents the universal quantifier.

## 2  Related Work

**Adversarial attacks:** Adversarial examples were first found in [59]. Since then, tremendous work have been done exploring the origins of this intriguing property of deep learning [25, 38, 20, 60, 22, 72] as well as designing more powerful attacks [23, 47, 43, 41, 9, 11] under various attack settings. [5] identified the gradient masking problem and showed that many defense methods could be broken with a few changes on the attacker. [13] proposed gradient-free black-box attacks and [32, 33, 12] further improved its efficiency. Recently, [34, 35] pointed out that adversarial examples are generated from the non-robust or invariant features hidden in the training data.

**Defensive adversarial learning:** Many defense approaches have been proposed to directly learn a robust model that can defend against adversarial attacks. [41] proposed a general framework of robust training by solving a min-max optimization problem. [62] proposed a new criterion to evaluate the convergence quality quantitatively. [70] theoretically studied the trade-off between natural accuracy and robust accuracy for adversarially trained models. [63] followed this framework and further improved its robustness by differentiating correctly classified and misclassified examples. [14] solve the problem by restricting the variation of outputs with respect to the inputs. [15, 54, 39] developed provably robust adversarial learning methods that have the theoretical guarantees on robustness. Recent works in [65, 50] focus on creating adversarial robust networks with faster training protocol. Another line of works focuses on increasing the effective size of the training data, either by pre-trained models [30] or by semi-supervised learning [10, 1, 44]. Very recently, [66] proposed to conduct adversarial weight perturbation aside from input perturbation to obtain more robust models. [24] achieves further robust models by practical techniques like weight averaging.

**Robustness and generalization:** Earlier works like [23] found that adversarial learning can reduce overfitting and help generalization. However, as the arms race between attackers and defenses keeps going, it is observed that strong adversarial attacks can cause severe damage to the model's natural accuracy [41, 70]. Many works [70, 61, 52, 19] attempt to explain this trade-off between robustness and natural generalization, while some other works proposed different perspectives. [56] confirmed that more training data has the potential to close this gap. [6] suggested that a robust model is computationally difficult to learn and optimize. [72] showed that there is still a large gap between the currently achieved model robustness and the theoretically achievable robustness limit on natural image distributions. [2] showed that the adversarial examples stem from the accumulation of small dense mixtures in the hidden weights during training and adversarial training works by removing such mixtures. Very recently, [51] showed that this trade-off stems from overparameterization and insufficient data in the linear regression setting. [68] proved that both accuracy and robustness are achievable through locally Lipschitz functions with separated data, and the gap between theory and practice is due to either failure to impose local Lipschitzness or insufficient generalization. [7] also studied the relationship between robustness and network size. In particular, [7] shows that overparametrization is necessary for robustness on two-layer neural networks, while we show that when networks get wider, they will have worse perturbation stability, and therefore larger regularization is needed to achieve better robustness.

## 3  Empirical Study on Network Width and Adversarial Robustness

In this section, we empirically study the relation between network width and robustness by first taking a closer look at the robust accuracy and the associated robust examples.

### 3.1  Characterization of Robust Examples

Robust accuracy is the standard evaluation metric of robustness, which measures the ratio of robust examples, i.e., examples that can still be correctly classified after adversarial attacks.

Previous empirical results suggest that wide models enjoy both better generalization ability and model robustness. Specifically, [41] proposed to extend ResNet [28] architecture to WideResNet [69] with a widen factor 10 for adversarial training on the CIFAR10 dataset and found that the increased model capacity significantly improves both robust accuracy and natural accuracy. Later works [70, 63] follow this finding and report their best result using the wide networks.

However, as shown by our findings in Figure 1, wider models actually lead to worse robust regularization effects, suggesting that wider models are not better in all aspects, and the relation between model robustness and network width may be more intricate than what people understood previously. To understand the intrinsic relationship between model robustness and network width, let us first take a closer look at the robust examples. Mathematically, robust examples can be defined as $\mathcal{S}_{\text{rob}} := \big\{ \mathbf{x} : \forall \widehat{\mathbf{x}} \in \mathbb{B}(\mathbf{x}, \epsilon), f(\boldsymbol{\theta}; \widehat{\mathbf{x}}) = y \big\}$. Note that by definition of robust examples, we have the following equation holds:

$$\underbrace{\big\{ \mathbf{x} : \forall \widehat{\mathbf{x}} \in \mathbb{B}(\mathbf{x}, \epsilon), f(\boldsymbol{\theta}; \widehat{\mathbf{x}}) = y \big\}}_{\text{robust examples}: \mathcal{S}_{\text{rob}}} = \underbrace{\big\{ \mathbf{x} : f(\boldsymbol{\theta}; \mathbf{x}) = y \big\}}_{\text{correctly classified examples}: \mathcal{S}_{\text{correct}}} \wedge \underbrace{\big\{ \mathbf{x} : \forall \widehat{\mathbf{x}} \in \mathbb{B}(\mathbf{x}, \epsilon), f(\boldsymbol{\theta}; \mathbf{x}) = f(\boldsymbol{\theta}; \widehat{\mathbf{x}}) \big\}}_{\text{stable examples}: \mathcal{S}_{\text{stable}}},$$

(3.1)

where $\wedge$ is the logical conjunction operator. (3.1) suggests that the robust examples are the intersection of two other sets: the correctly classified examples (examples whose predictions are the correct labels) and the stable examples (examples whose predictions are the same within the $\ell_p$ norm ball). A more direct illustration of this relationship can be found in Figure 2. While the natural accuracy measures the ratio of correctly classified examples $|\mathcal{S}_{\text{correct}}|$ against the whole sample set, to our knowledge, there does not exist a metric measuring the ratio of stable examples $|\mathcal{S}_{\text{stable}}|$ against whole the sample set. Here we formally define this ratio as the *perturbation stability*, which measures the fraction of examples whose predictions cannot be perturbed as reflected in the robust regularization term in (1.1).

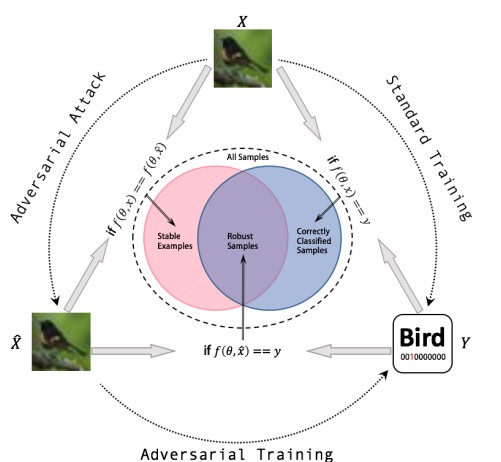

Figure 2: An illustration of robust, correctly classified, and stable examples in (3.1).

### 3.2   Evaluation of Perturbation Stability

We apply the TRADES [70] method, which is one of the strongest baselines in robust training, on CIFAR10 dataset and plot the robust accuracy, natural accuracy, and perturbation stability against the training epochs in Figure 3. Experiments are conducted on WideResNet-34 [69] with various widen factors. For each network, when robust accuracy reaches the highest point, we record all three metrics and show their changing trend against network width in Figure 3(d). From Figure 3(d), we can observe that the perturbation stability decreases monotonically as the network width increases. This suggests that wider models are actually more vulnerable to adversarial perturbation. In this sense, the increased network width could hurt the overall model robustness to a certain extent. This can be seen from Figure 3(d), where the robust accuracy of widen-factor 5 is actually slightly better than that of widen-factor 10.

Aside from the relation with model width, we can also gain other insights from perturbation stability:

1. Unlike robust accuracy and natural accuracy, perturbation stability gradually gets worse during the training process. This makes sense since an untrained model that always outputs the same label will have perfect stability, and the training process tends to break this perfect stability. From another perspective, the role of robust regularization in (1.1) is to encourage perturbation stability, such that the model predictions remain the same under small perturbations, which in turn improves model robustness.

2. Previous works [70, 61, 52] have argued that there exists a trade-off between natural accuracy and robust accuracy. However, from (3.1), we can see that robust accuracy and natural accuracy are coupled with each other, as a robust example must first be correctly classified. When the

natural accuracy goes to zero, the robust accuracy will become zero. On the other hand, higher natural accuracy also implies that more examples will likely become robust examples. Works including [51] and [45] also challenged this robust-natural trade-off [61] does not hold for some cases. Therefore, we argue that the real trade-off here should be between natural accuracy and perturbation stability and the robust accuracy is actually the consequence of this trade-off.

3. [53] has recently shown that adversarial training suffers from over-fitting as the robust accuracy might get worse as training proceeds, which can be seen in Figure 3(a). We found that the origin of this over-fitting is mainly attributed to the degenerate perturbation stability (Figure 3(c)) rather than the natural risk (Figure 3(b)). Future works of adversarial training may consider evaluating our perturbation stability to understand how their method takes effects. Do they only help natural risk, or robust regularization, or maybe both of them.

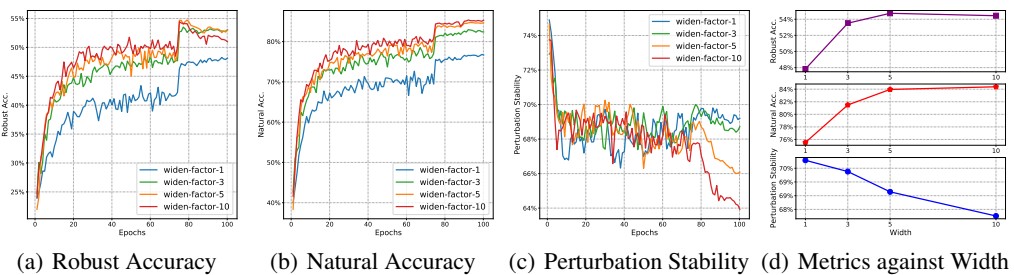

(a) Robust Accuracy     (b) Natural Accuracy     (c) Perturbation Stability   (d) Metrics against Width

Figure 3: Plots of (a) robust accuracy, (b) natural accuracy, and (c) perturbation stability against training epochs for networks of different width. Results are acquired on CIFAR10 with the adversarial training method TRADES and architectures of WideResNet-34. Training schedule is the same as the original work [70]. We record all three metrics when robust accuracy reaches the highest point and plot them against network width in (d).

# 4 Why Larger Network Width Leads to Worse Perturbation Stability?

Our empirical findings in Section 3 explains why the larger network width may not help model robustness as it leads to worse perturbation stability. However, it still remains unclear what is the underlying reasons for the negative correlation between the perturbation stability and the model width. In this section, we show that larger network width naturally leads to worse perturbation stability from a theoretical perspective. Specifically, we first relate perturbation stability with the network's local Lipschitzness and then study the relationship between local Lipschitzness and the model width by leveraging recent studies on neural tangent kernels [36, 3, 8, 73, 21].

## 4.1 Perturbation Stability and Local Lipschitzness

Previous works [29, 64] usually relate local Lipschitzness with network robustness, suggesting that smaller local Lipschitzness leads to robust models. Here we show that local Lipshctzness is more directly linked to perturbation stability, through which it further influences model robustness.

As a start, let us first recall the definition of Lipschitz continuity and its relation with gradient norms.

**Lemma 4.1** (Lipschitz continuity and gradient norm [49]). Let $\mathcal{D} \in \mathbb{R}^d$ denotes a convex compact set, $f$ is a Lipschitz function if for all $\mathbf{x}, \mathbf{x}' \in \mathcal{D}$, it satisfies

$$|f(\mathbf{x}') - f(\mathbf{x})| \leq L\|\mathbf{x}' - \mathbf{x}\|_p,$$

where $L = \sup_{\mathbf{x} \in \mathcal{D}}\{\|\nabla f(\mathbf{x})\|_q\}$ and $1/p + 1/q = 1$.

Intuitively speaking, Lipschitz continuity guarantees that small perturbation in the input will not lead to large changes in the function output. In the adversarial training setting where the perturbation $\mathbf{x}'$ can only be chosen within the neighborhood of $\mathbf{x}$, we focus on the local Lipschitz constant where we restrict $\mathbf{x}' \in \mathbb{B}(\mathbf{x}, \epsilon)$ and $L = \sup_{\mathbf{x}' \in \mathbb{B}(\mathbf{x}, \epsilon)}\{\|\nabla f(\mathbf{x}')\|_q\}$.

Now suppose our neural network loss function is local Lipschitz, let $\mathbf{x}'$ be our computed adversarial example $\widehat{\mathbf{x}}$ and $\mathbf{x}$ be the original example, the robust regularization term satisfies

$$\max_{\widehat{\mathbf{x}} \in \mathbb{B}(\mathbf{x}, \epsilon)} \left[ \mathcal{L}(\boldsymbol{\theta}; \widehat{\mathbf{x}}, y) - \mathcal{L}(\boldsymbol{\theta}; \mathbf{x}, y) \right] \leq L \max_{\widehat{\mathbf{x}} \in \mathbb{B}(\mathbf{x}, \epsilon)} \left[ \|\widehat{\mathbf{x}} - \mathbf{x}\|_p \right] \leq \epsilon L, \tag{4.1}$$

where the first inequality is due to local Lipschitz continuity and $L = \sup_{\mathbf{x}' \in \mathbb{B}(\mathbf{x}, \epsilon)} \{ \|\nabla \mathcal{L}(\boldsymbol{\theta}; \mathbf{x}', y)\|_q \}$. (4.1) shows that the local Lipschitz constant is directly related to the robust regularization term, which can be used as a surrogate loss for the perturbation stability.

## 4.2 Local Lipschitzness and Network Width

Now we study how the network width affects the perturbation stability via studying the local Lipschitz constant.

Recently, a line of research emerges, which tries to theoretically understand the optimization and generalization behaviors of over-parameterized deep neural networks through the lens of the neural tangent kernel (NTK) [36, 3, 8, 73]. By showing the equivalence between over-parameterized neural networks and NTK in the finite width setting, this type of analysis characterizes the optimization and generalization performance of deep learning by the network architecture (e.g., network width, which we are particularly interested in). Recently, [21] also analyzed the convergence of adversarial training for over-parameterized neural networks using NTK. Here, we will show that the local Lipschitz constant increases with the model width.

In specific, let $m$ be the network width and $H$ be the network depth. Define an $H$-layer fully connected neural network as follows

$$f(\mathbf{x}) = \mathbf{a}^\top \sigma(\mathbf{W}^{(H)} \sigma(\mathbf{W}^{(H-1)} \cdots \sigma(\mathbf{W}^{(1)} \mathbf{x}) \cdots)),$$

where $\mathbf{W}^{(1)} \in \mathbb{R}^{m \times d}$, $\mathbf{W}^{(h)} \in \mathbb{R}^{m \times m}, h = 2, \ldots, H$ are the weight matrices, $\mathbf{a} \in \mathbb{R}^m$ is the output layer weight vector, and $\sigma(\cdot)$ is the entry-wise ReLU activation function. For notational simplicity, we denote by $\mathbf{W} = \{\mathbf{W}^{(H)}, \ldots, \mathbf{W}^{(1)}\}$ the collection of weight matrices and by $\mathbf{W}_0 = \{\mathbf{W}_0^{(H)}, \ldots, \mathbf{W}_0^{(1)}\}$ the collection of initial weight matrices. Following [21], we assume the first layer and the last layer's weights are fixed, and $\mathbf{W}$ is updated via projected gradient descent with projection set $B(R) = \{\mathbf{W} : \|\mathbf{W}^{(h)} - \mathbf{W}_0^{(h)}\|_F \leq R/\sqrt{m}, h = 1, 2, \ldots, H\}$. We have the following lemma upper bounding the input gradient norm.

**Lemma 4.2.** For any given input $\mathbf{x} \in \mathbb{R}^d$ and $\ell_2$ norm perturbation limit $\epsilon$, if $m \geq \max(d, \Omega(H \log(H)))$, $R/\sqrt{m} + \epsilon \leq c/(H^6 (\log m)^3)$ for some sufficient small $c > 0$, then with probability at least $1 - O(H) e^{-\Omega(m(R/\sqrt{m}+\epsilon)^{2/3} H)}$, we have for any $\mathbf{x}' \in \mathbb{B}(\mathbf{x}, \epsilon)$ and Lipschitz loss $\mathcal{L}$, the input gradient norm satisfies

$$\|\nabla \mathcal{L}(f(\mathbf{x}'), y)\|_2 = O(\sqrt{mH}).$$

The proof of Lemma 4.2 can be found in the supplemental materials. Note that Lemma 4.2 holds for any $\mathbf{x}' \in \mathbb{B}(\mathbf{x}, \epsilon)$, therefore, the maximum input gradient norm in the $\epsilon$-ball is also in the order of $O(\sqrt{mH})$. Lemma 4.2 suggests that the local Lipschitz constant is closely related to the neural network width $m$. In particular, the local Lipschitz constant scales as the square root of the network width. This in theory explains why wider networks are more vulnerable to adversarial perturbation.

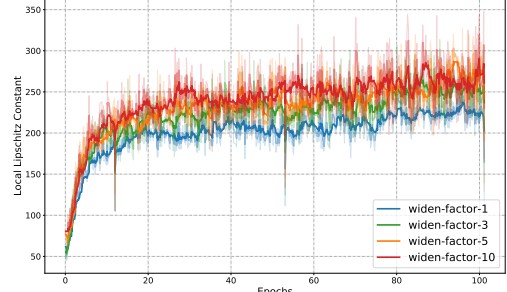

Figure 4: Plot of approximated local Lipschitz constant along the adversarial training trajectory. Models are trained by TRADES [70] on CIFAR10 dataset using WideResNet model. Wider networks in general have larger local Lipschitz constants.

In order to further verify the above theoretical result, we empirically calculate the local Lipschitz constant. In detail, for commonly used $\ell_\infty$ norm threat model, we evaluate the quantity $\sup_{\mathbf{x}' \in \mathbb{B}(\mathbf{x}, \epsilon)} \{ \|\nabla \mathcal{L}(\boldsymbol{\theta}; \mathbf{x}', y)\|_1 \}$ along the adversarial training trajectory for networks with different widths. Note that solving this maximization problem along the entire training trajectory is computationally expensive or even intractable.

Table 1: The three metrics under PGD attack with different $\lambda$ on CIFAR10 dataset using WideResNet-34 model. We test TRADES as well as our (generalized) adversarial training. Each experiment is repeated three times. The highest robustness value for each column is annotated with bold number. From the table, we can tell that: 1) The best choice of $\lambda$ increases as the network width increases; 2) For models with the same width, the larger $\lambda$ always leads to higher perturbation stability; 3) With the same $\lambda$, the larger width always hurts perturbation stability, which backs up our claim in Section 4.2.

| $\lambda$ | Robust Accuracy (%) | | | Natural Accuracy (%) | | | Perturbation Stability (%) | | |
|---|---|---|---|---|---|---|---|---|---|
| | width-1 | width-5 | width-10 | width-1 | width-5 | width-10 | width-1 | width-5 | width-10 |
| | | | | TRADES [70] | | | | | |
| 6 | 47.81±.09 | 54.45±.16 | 54.18±.39 | **76.26±.10** | **84.44±.06** | **84.90±.80** | 69.33±.05 | 68.27±.22 | 67.25±.39 |
| 9 | **48.01±.06** | 55.34±.17 | 55.29±.45 | 73.78±.30 | 82.77±.07 | 84.13±.28 | 71.92±.33 | 70.66±.26 | 69.08±.80 |
| 12 | 47.87±.06 | **55.61±.04** | 55.98±.13 | 72.29±.25 | 81.59±.20 | 83.59±.62 | 73.33±.16 | 72.00±.20 | 70.18±.67 |
| 15 | 47.15±.13 | 55.49±.15 | 55.96±.09 | 70.98±.24 | 80.69±.08 | 82.81±.19 | 73.79±.27 | 72.87±.03 | 70.87±.23 |
| 18 | 47.02±.13 | 55.43±.12 | **56.43±.17** | 70.13±.06 | 79.97±.12 | 82.21±.21 | 74.63±.11 | 73.77±.13 | 72.04±.30 |
| 21 | 46.26±.19 | 55.31±.20 | 56.07±.21 | 68.95±.38 | 79.25±.23 | 81.74±.12 | **75.17±.28** | **74.15±.38** | **72.11±.12** |
| | | | | Adversarial Training [41] | | | | | |
| 1.00 | 47.99±.16 | 50.87±.42 | 50.12±.13 | **77.30±.01** | **85.82±.01** | 85.62±.81 | 66.48±.24 | 62.23±.42 | 61.62±.46 |
| 1.25 | **49.24±.12** | 53.10±.09 | 51.97±.46 | 74.04±.47 | 84.73±.22 | **86.25±.12** | 70.34±.54 | 65.24±.08 | 62.94±.35 |
| 1.50 | 49.11±.03 | 54.15±.03 | 53.25±.52 | 72.16±.25 | 84.35±.19 | 85.50±.57 | 72.10±.11 | 66.65±.06 | 64.51±.72 |
| 1.75 | 48.32±.63 | **54.36±.14** | 53.65±.80 | 70.66±.46 | 83.95±.30 | 85.52±.24 | 72.43±.40 | 67.31±.03 | 65.67±.10 |
| 2.00 | 47.44±.06 | 54.10±.15 | **55.78±.22** | 69.67±.09 | 83.49±.06 | 85.41±.13 | **72.73±.04** | **67.53±.01** | **65.71±.15** |

Therefore, we approximate this quantity by choosing the maximum input gradient $\ell_1$-norm among the 10 attack steps for each iteration. Figure 4 shows that larger network width indeed leads to larger local Lipschitz constant values. This backup the theoretical results in Lemma 4.2.

# 5 Experiments

From Section 4, we know that wider networks have worse perturbation stability. This suggests that to fully unleash the potential of wide model architectures, we need to carefully control the decreasing of the perturbation stability on wide models. One natural strategy to do this is by adopting a larger robust regularization parameter $\lambda$ in (1.1). In this section, we conduct thorough experiments to verify whether this strategy can mitigate the negative effects on perturbation stability and achieve better performances for wider networks.

It is worth noting that due to the high computational overhead of adversarial training on wide networks, previous works [70] tuned $\lambda$ on smaller networks (ResNet18 [27]) and directly apply it on wider ones, neglecting the influence of model capacity. Our analysis suggests that using the same $\lambda$ for models with different widths is suboptimal, and one should use a larger $\lambda$ for wider models in order to get better model robustness.

## 5.1 Experimental Settings

We conduct our experiments on CIFAR10 [37] dataset, which is the most popular dataset in the adversarial training literature. It contains images from 10 different categories, with $50k$ images for training and $10k$ for testing. Here we first conduct our experiments using the TRADES [70] method. Networks are chosen from WideResNet [69] with different widen factor from $1, 5, 10$. The batch size is set to 128, and we train each model for 100 epochs. The initial learning rate is set to be 0.1. We adopt a slightly different learning rate decay schedule: instead of dividing the learning rate by 10 after 75-th epoch and 90-th epoch as in [41, 70, 63], we halve the learning rate for every epoch after the 75-th epoch, for the purpose of preventing over-fitting. For evaluating the model robustness, we perform the standard PGD attack [41] using 20 steps with step size 0.007, and $\epsilon = 8/255$. Note that previous works [70, 63] report their results using step size 0.003, which we found is actually less effective than ours. All experiments are conducted on a single NVIDIA V100 GPU.

## 5.2 Model Robustness with Larger Robust Regularization Parameter

We first compare the robustness performance of models with different network width using robust regularization parameters chosen from $\{6, 9, 12, 15, 18, 21\}$ for TRADES [70]. Results of different evaluation metrics are presented in Table 1.

Table 2: Robust accuracy (%) for different datasets, architectures and regularization parameters under various attacks. The highest results are evaluated for three times of randomly started attack. Our approach of boosting regularization for wider models apply to all cases. The value of $w$ and $k$ represents the network width.

| Dataset | Architecture | widen-factor/ growth-rate | regulari- zation | PGD | C&W | FAB | Square |
|---|---|---|---|---|---|---|---|
| CIFAR10 | WideResNet-34 | $w=1$ | $\lambda=6$ | **47.92±.01** | **44.95±.03** | **44.31±.04** | **49.25±.02** |
| | | | $\lambda=12$ | 47.91±.04 | 44.24±.02 | 43.71±.05 | 47.75±.02 |
| | | | $\lambda=18$ | 46.92±.05 | 43.48±.03 | 43.00±.01 | 46.01±.05 |
| | | $w=5$ | $\lambda=6$ | 54.50±.03 | 53.14±.03 | 52.13±.05 | 56.79±.02 |
| | | | $\lambda=12$ | **55.56±.04** | **53.28±.04** | **52.55±.02** | **56.88±.05** |
| | | | $\lambda=18$ | 55.21±.02 | 52.64±.02 | 52.18±.01 | 56.31±.01 |
| | | $w=10$ | $\lambda=6$ | 54.23±.04 | 54.02±.03 | 52.68±.07 | 57.64±.03 |
| | | | $\lambda=12$ | 55.80±.06 | 54.41±.01 | 53.57±.04 | 57.72±.10 |
| | | | $\lambda=18$ | **56.29±.10** | **54.57±.02** | **54.06±.02** | **58.04±.05** |
| | DenseNet-BC-40 | $k=12$ | $\lambda=6$ | **44.79±.02** | 40.83±.03 | **40.07±.03** | **45.66±.05** |
| | | | $\lambda=12$ | 44.66±.03 | **40.91±.03** | 39.88±.01 | 44.23±.04 |
| | | | $\lambda=18$ | 44.38±.05 | 40.63±.03 | 39.42±.01 | 43.31±.04 |
| | | $k=64$ | $\lambda=6$ | 55.51±.01 | 52.76±.04 | 51.74±.02 | 57.24±.01 |
| | | | $\lambda=12$ | **55.85±.03** | **52.98±.02** | **52.10±.03** | **57.34±.04** |
| | | | $\lambda=18$ | 55.71±.03 | 52.83±.06 | 51.66±.04 | 55.21±.03 |
| CIFAR100 | WideResNet-34 | $w=1$ | $\lambda=6$ | **24.28±.02** | **20.24±.01** | **19.97±.02** | **22.91±.02** |
| | | | $\lambda=12$ | 24.18±.04 | 20.15±.02 | 19.83±.01 | 22.78±.01 |
| | | | $\lambda=18$ | 23.99±.03 | 20.01±.02 | 19.01±.01 | 22.04±.01 |
| | | $w=5$ | $\lambda=6$ | 30.73±.03 | 27.25±.05 | 26.01±.03 | 30.11±.03 |
| | | | $\lambda=12$ | **31.57±.02** | **27.83±.02** | **27.08±.01** | **30.45±.01** |
| | | | $\lambda=18$ | 31.38±.01 | 27.66±.04 | 26.94±.03 | 30.02±.01 |
| | | $w=10$ | $\lambda=6$ | 30.48±.02 | 27.98±.01 | 27.00±.11 | 30.45±.06 |
| | | | $\lambda=12$ | 31.75±.09 | 29.25±.04 | 28.14±.03 | 31.23±.04 |
| | | | $\lambda=18$ | **32.98±.03** | **29.83±.01** | **28.78±.02** | **32.02±.01** |

From Table 1, we can observe that the best robust accuracy for width-1 network is achieved when $\lambda = 9$, yet for width-5 network, the best robust accuracy is achieved when $\lambda = 12$, and for width-10 network, the best $\lambda$ is 18. This suggests that wider networks indeed need a larger robust regularization parameter to unleash the power of wide model architecture fully. Our exploration also suggests that the optimal choice of $\lambda$ for width-10 network is 18 under the same setting as [70], which is three times larger than the one used in the original paper, leading to an average improvement of 2.25% on robust accuracy. It is also worth noting that enlarging $\lambda$ indeed leads to improved perturbation stability. Under the same $\lambda$, wider networks have worse perturbation stability. This observation is rather consistent with our empirical and theoretical findings in Sections 3 and 4. As stated in Section 3.2, the real trade-off is between natural accuracy and perturbation stability rather than robust accuracy. Also, the stability provides a clear hint for finding the best choice of $\lambda$.

We further show that our strategy also applies to the original adversarial training [41], as shown by the bottom part of Table 1. Proper adaptations should be made to boost the robust regularization for original (generalized) adversarial training. We show the detail of the adaptations in the Appendix. As shown by the table, the large improvements on both TRADES and adversarial training using our boosting strategy suggest that adopting larger $\lambda$ is crucial in unleashing the full potential of wide models, which is usually neglected in practice.

## 5.3 Experiments on Different Datasets and Architectures

To show that our theory is universal and is applicable to various datasets and architectures, we conduct extra experiments on the CIFAR100 dataset and DenseNet model [31]. For the DenseNet models, the growth rate $k$ denotes how fast the number of channels grows and thus becomes a suitable measure of network width. Following the original paper [31], we choose DenseNet-BC-40 and use models with different growth rates to verify our theory.

Experimental results are shown in Table 2. For completeness, we also report the results under four different attack methods and settings, including PGD [41], C&W [9], FAB [16], and Square [4]. We

**Algorithm 1** Width Adjusted Regularization

1: **Input**: initial weights $\boldsymbol{\theta}_0$, WAR parameter $\zeta$, learning rate $\eta$, adversarial attack $\mathcal{A}$
2: $\lambda_0 = 0, \alpha = 0.1$
3: **for** $t = 1, \ldots, T$ **do**
4:     Get mini-batch $\{(\mathbf{x}_1, y_1), \ldots, (\mathbf{x}_m, y_m)\}$
5:     **for** $i = 1, \ldots, m$ (in parallel) **do**
6:         $\widehat{\mathbf{x}}_i \leftarrow \mathcal{A}(\mathbf{x}_i)$
7:         $l_{\text{nat}} \leftarrow \mathcal{L}(\boldsymbol{\theta}_t; \mathbf{x}_i, y_i)$
8:         $l_{\text{rob}} \leftarrow \mathcal{L}(\boldsymbol{\theta}_t; \widehat{\mathbf{x}}_i, y_i) - \mathcal{L}(\boldsymbol{\theta}_t; \mathbf{x}_i, y_i)$
9:         $\lambda_t \leftarrow \max(\lambda_{t-1} + \alpha \cdot (\zeta - (l_{\text{nat}}/l_{\text{rob}}), 0)$
10:        $\boldsymbol{\theta}_t \leftarrow \boldsymbol{\theta}_{t-1} - (\eta/m) \sum_{i=1}^{m} \nabla_{\boldsymbol{\theta}} [l_{\text{nat}} + \lambda_t \cdot l_{\text{rob}}]$
11:     **end for**
12: **end for**

Table 3: Comparison of TRADES with different tuning strategies. N/A denotes no fine-tuning of the current model (tuning on small networks only). Manual represents exhaustive fine-tuning.

| Model | Tuning | $\lambda$ | PGD | GPU hours |
|---|---|---|---|---|
| WRN-34-1 | N/A | 6.00 | 47.81 | 12+18=30 |
| | Manual | 9.00 | 48.01 | 12×6=72 |
| | WAR | 9.12 | **48.06** | 12+18=30 |
| WRN-34-5 | N/A | 6.00 | 54.45 | 20+18=38 |
| | Manual | 12.00 | 55.61 | 20×6=120 |
| | WAR | 14.37 | **55.62** | 20+18=38 |
| WRN-34-10 | N/A | 6.00 | 54.18 | 32+18=50 |
| | Manual | 18.00 | 56.43 | 32×6=192 |
| | WAR | 16.43 | **56.46** | 32+18=50 |

adopt the best $\lambda$ from Table 1 and show the corresponding performance on models with different widths. It can be seen that our strategy of using a larger robust regularization parameter works very well across different datasets and networks. On the WideResNet model, we observe clear patterns as in Section 5.2. On the DenseNet model, although the best regularization $\lambda$ is different from that of WideResNet, wider models, in general, still require larger $\lambda$ for better robustness. On CIFAR100, our strategy raises the standard PGD score of the widest model from 30.48% to 32.98%.

## 5.4 Width Adjusted Regularization

Our previous analysis has shown that larger model width may hurt adversarial robustness without properly choosing the regularization parameter $\lambda$. However, exhaustively cross-validating $\lambda$ on wider networks can be extremely time-consuming in practice. To address this issue, we investigate the possibility of automatically adjusting $\lambda$ according to the model width, based on our existing knowledge obtained in fine-tuning smaller networks, which is much cheaper. Note that the key to achieving the best robustness is to well balance between the natural risk term and the robust regularization term in (1.1). Although the regularization parameter $\lambda$ cannot be directly applied from thinner networks to wider networks (as suggested by our analyses), the best ratio between the natural risk and the robust regularization across different width models can be kept roughly the same. Following this idea, we design the **W**idth **A**djusted **R**egularization (WAR) method, which is summarized in Algorithm 1. Specifically, we first manually tune the best $\lambda$ for a thin network and record the ratio $\zeta$ between the natural risk and the robust regularization when the training converges. Then, on training wider networks, we adaptively[2] adjust $\lambda$ to encourage the ratio between the natural risk and the robust regularization to stay close to $\zeta$. Let's take an example here. We first cross-validate $\lambda$ on a thin network with widen factor 0.5 and identify the best $\lambda = 6$ and $\zeta = 30$ with 18 GPU hours in total. Now we compare three different strategies for training wider models and summarize the results in Table 3: 1) directly apply $\lambda = 6$ with no fine-tuning on the current model; 2) exhaustive manual fine-tuning from $\lambda = 6.0$ to $\lambda = 21.0$ (6 trials) as in Table 1; 3) our WAR strategy. Table 3 shows that the final $\lambda$ generated by WAR on wider models are consistent with the exhaustively tuned best $\lambda$. Compared to the exhaustive manual tuning strategy, WAR achieves even slightly better model robustness with much less overall training time ($\sim$4 times speedup for WRN-34-10 model). On the other hand, directly using $\lambda = 6$ with no tuning on the wide models leads to much worse model robustness while having the same overall training time. This verifies the effectiveness of our proposed WAR method.

## 5.5 Comparison of Robustness on Wide Models

Previous experiments in Section 5.2 and Section 5.3 have shown the effectiveness of our proposed strategy on using larger robust regularization parameter for wider models. In order to ensure that this strategy does not lead to any obfuscated gradient problem [5] and gives a false sense of robustness, we further conduct experiments using stronger attacks. In particular, we choose to evaluate our best models on the AutoAttack algorithm [17], which is an ensemble attack method that contains four different white-box and black-box attacks for the best attack performances.

---

[2] the learning rate $\alpha$ for $\lambda_t$ in Algorithm 1 is not sensitive and needs no extra tuning.

We evaluate models trained with WAR, with or without extra unlabeled data [10], and report the robust accuracy in Table 4. Note that the results of other baselines are directly obtained from the AutoAttack leaderboard[3]. From Table 4, we can see that our WAR significantly improves the baseline TRADES models on WideResNet. This experiment further verifies the effectiveness of our proposed strategy.

## 6 Conclusions

In this paper, we studied the relation between network width and adversarial robustness in adversarial training, a principled approach to train robust neural networks. We showed that the model robustness is closely related to both natural accuracy and perturbation stability, while the balance between the two is controlled by the robust regularization parameter $\lambda$. With the same value of $\lambda$, the natural accuracy is better on wider models while the perturbation stability actually becomes worse, leading to a possible decrease in the overall model robustness. We showed the origin of this problem by relating perturbation stability with local Lipschitzness and leveraging recent studies on the neural tangent kernel to prove that larger network width leads to worse perturbation

Table 4: Robust accuracy (%) comparison on CIFAR10 under AutoAttack. † indicates training with extra unlabeled data.

| Methods | Model | AutoAttack |
|---|---|---|
| TRADES [70] | WRN-34-10 | 53.08 |
| Early-Stop [53] | WRN-34-20 | 53.42 |
| FAT [71] | WRN-34-10 | 53.51 |
| HE [46] | WRN-34-20 | 53.74 |
| WAR | WRN-34-10 | **54.73** |
| MART [63]† | WRN-28-10 | 56.29 |
| HYDRA [57]† | WRN-28-10 | 57.14 |
| RST [10]† | WRN-28-10 | 59.53 |
| WAR† | WRN-28-10 | **60.02** |
| WAR† | WRN-28-20 | **61.84** |

stability. Our analyses suggest that: 1) proper tuning of $\lambda$ on wider models is necessary despite being extremely time-consuming; 2) practitioners should adopt a larger $\lambda$ for training wider networks. Finally, we propose the Width Adjusted Regularization, which significantly saves the tuning time for robust training on wide models.

## 7 Acknowledgments

We thank the anonymous reviewers for their helpful comments. Boxi Wu, Deng Cai and Xiaofei He are supported in part by The National Key Research and Development Program of China (Grant Nos: 2018AAA0101400), The National Nature Science Foundation of China (Grant Nos: 62036009, U1909203, 61936006), and Innovation Capability Support Program of Shaanxi (Program No. 2021TD-05).

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
