# A Proof of Lemma 4.2

**Lemma A.1** (Restatement of Lemma 4.2). For any given input $\mathbf{x} \in \mathbb{R}^d$ and $\ell_2$ norm perturbation limit $\epsilon$, if $m \geq \max(d, \Omega(H \log(H)))$, $\frac{R}{\sqrt{m}} + \epsilon \leq \frac{c}{H^6 (\log m)^3}$ for some sufficient small $c$, then with probability at least $1 - O(H)e^{-\Omega(m(R/\sqrt{m}+\epsilon)^{2/3}H)}$, we have for any $\mathbf{x}' \in \mathbb{B}(\mathbf{x}, \epsilon)$ and Lipschitz loss $\mathcal{L}$, the input gradient norm satisfies

$$\|\nabla \mathcal{L}(f(\mathbf{x}'), y)\|_2 = O\big(\sqrt{mH}\big).$$

*Proof.* The major part of this proof is inspired from [19]. Let $\mathbf{D}^{(h)}(\mathbf{W}, \mathbf{x}) = \text{diag}(\mathbb{1}\{\mathbf{W}^{(h)}\sigma(\cdots\sigma(\mathbf{W}^{(1)}\mathbf{x})) > 0\})$ be a diagonal sign matrix. Then the neural network function can be rewritten as follows:

$$f(\mathbf{x}) = \mathbf{a}^\top \mathbf{D}^{(H)}(\mathbf{W}, \mathbf{x})\mathbf{W}^{(H)} \cdots \mathbf{D}^{(1)}(\mathbf{W}, \mathbf{x})\mathbf{W}^{(1)}\mathbf{x}.$$

By the chain rule of the derivatives, the input gradient norm can be further written as

$$\begin{aligned}
\|\nabla \mathcal{L}(f(\mathbf{x}'), y)\|_2 &= \|\mathcal{L}'(f(\mathbf{x}'), y) \cdot \nabla f(\mathbf{x}')\|_2 \\
&\leq \|\mathcal{L}'(f(\mathbf{x}'), y)\|_2 \cdot \|\nabla f(\mathbf{x}')\|_2 \\
&= \|\mathcal{L}'(f(\mathbf{x}'), y)\|_2 \cdot \|\mathbf{a}^\top \mathbf{D}^{(H)}(\mathbf{W}, \mathbf{x}')\mathbf{W}^{(H)} \cdots \mathbf{D}^{(1)}(\mathbf{W}, \mathbf{x}')\mathbf{W}^{(1)}\|_2. \quad \text{(A.1)}
\end{aligned}$$

Now let us focus on the term $\|\mathbf{a}^\top \mathbf{D}^{(H)}(\mathbf{W}, \mathbf{x}')\mathbf{W}^{(H)} \cdots \mathbf{D}^{(1)}(\mathbf{W}, \mathbf{x}')\mathbf{W}^{(1)}\|_2$. Note that by triangle inequality,

$$\begin{aligned}
&\|\mathbf{a}^\top \mathbf{D}^{(H)}(\mathbf{W}, \mathbf{x}')\mathbf{W}^{(H)} \cdots \mathbf{D}^{(1)}(\mathbf{W}, \mathbf{x}')\mathbf{W}^{(1)}\|_2 \\
&\leq \|\mathbf{a}^\top \mathbf{D}^{(H)}(\mathbf{W}, \mathbf{x}')\mathbf{W}^{(H)} \cdots \mathbf{D}^{(1)}(\mathbf{W}, \mathbf{x}')\mathbf{W}^{(1)} - \mathbf{a}^\top \mathbf{D}^{(H)}(\mathbf{W}_0, \mathbf{x})\mathbf{W}_0^{(H)} \cdots \mathbf{D}^{(1)}(\mathbf{W}_0, \mathbf{x})\mathbf{W}_0^{(1)}\|_2 \\
&\quad + \|\mathbf{a}^\top \mathbf{D}^{(H)}(\mathbf{W}_0, \mathbf{x})\mathbf{W}_0^{(H)} \cdots \mathbf{D}^{(1)}(\mathbf{W}_0, \mathbf{x})\mathbf{W}_0^{(1)}\|_2. \quad \text{(A.2)}
\end{aligned}$$

Note that $\mathbf{W}$ is updated via projected gradient descent with projection set $B(R)$. Therefore, by Equation (12) in Lemma A.5 of [19] we have

$$\begin{aligned}
&\|\mathbf{a}^\top \mathbf{D}^{(H)}(\mathbf{W}, \mathbf{x}')\mathbf{W}^{(H)} \cdots \mathbf{D}^{(1)}(\mathbf{W}, \mathbf{x}')\mathbf{W}^{(1)} - \mathbf{a}^\top \mathbf{D}^{(H)}(\mathbf{W}_0, \mathbf{x})\mathbf{W}_0^{(H)} \cdots \mathbf{D}^{(1)}(\mathbf{W}_0, \mathbf{x})\mathbf{W}_0^{(1)}\|_2 \\
&= O\bigg(\Big(\frac{R}{\sqrt{m}} + \epsilon\Big)^{1/3} H^2 \sqrt{m \log m}\bigg), \quad \text{(A.3)}
\end{aligned}$$

and by Lemma A.3 in [19] we have

$$\|\mathbf{a}^\top \mathbf{D}^{(H)}(\mathbf{W}_0, \mathbf{x})\mathbf{W}_0^{(H)} \cdots \mathbf{D}^{(1)}(\mathbf{W}_0, \mathbf{x})\mathbf{W}_0^{(1)}\|_2 = O(\sqrt{mH}). \quad \text{(A.4)}$$

Combining (A.2), (A.3), (A.4), when $\frac{R}{\sqrt{m}} + \epsilon \leq \frac{c}{H^6 (\log m)^3}$, we have

$$\|\mathbf{a}^\top \mathbf{D}^{(H)}(\mathbf{W}, \mathbf{x}')\mathbf{W}^{(H)} \cdots \mathbf{D}^{(1)}(\mathbf{W}, \mathbf{x}')\mathbf{W}^{(1)}\|_2 = O(\sqrt{mH}). \quad \text{(A.5)}$$

By substituting (A.5) into (A.1) we have,

$$\|\nabla \mathcal{L}(f(\mathbf{x}'), y)\|_2 \leq \|\mathcal{L}'(f(\mathbf{x}'), y)\|_2 \cdot \|\mathbf{a}^\top \mathbf{D}^{(H)}(\mathbf{W}, \mathbf{x}')\mathbf{W}^{(H)} \cdots \mathbf{D}^{(1)}(\mathbf{W}, \mathbf{x}')\mathbf{W}^{(1)}\|_2 = O(\sqrt{mH}),$$

where the last inequality holds since $\|\mathcal{L}'(f(\mathbf{x}'), y)\|_2 = O(1)$ due to the Lipschitz condition of loss $\mathcal{L}$. This concludes the proof. $\square$

# B The Experimental Detail for Reproducibility

All experiments are conducted on a single NVIDIA V100. It runs on the GNU Linux Debian 4.9 operating system. The experiment is implemented via PyTorch 1.6.0. We adopt the public released codes of PGD [39], TRADES [68], and RST [8] and adapt them for our own settings, including inspecting the loss value of robust regularization and the local Lipschitzness.

CIFAR100 contains 50k images for 100 classes, which means that it has much fewer images for each class compared with CIFAR10. This makes the learning problem of CIFAR100 much harder. For DenseNet architecture, we adopt the 40 layers model with the bottleneck design, which is the DenseNet-BC-40. It has three building blocks, with each one having the same number of layers. This is the same architecture tested in the original paper of DenseNet for CIFAR10. For simplicity reason, we make the training schedule stay the same with the one used for WideResNet, which is the decay learning rate schedule. As DenseNet gets deeper, its channel number (width) will be multiplied with the growing rate k. Thus, as k gets larger, the width of DenseNet also does. Although this mechanism slightly differs from the widen factor of WideResNet, which amplify all layers with the same ratio.

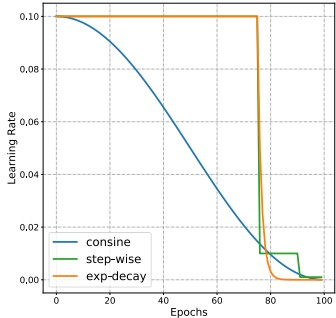

Figure 5: The changing trend leanring rate against training epochs for different learning rate schedule.

## C    The Exponential Decay Learning Rate

To demonstrate the fact that the over-fitting problem all comes from perturbation stability in Section 3.2(3), we use the training schedule of the original work for Figure 2. Aside from that, all the other experiments and plots are results under our proposed learning rate schedule, which halve the learning rate for every epochs after the 75-th epoch and can prevent over-fitting. Different learning rate schedules are shown in Figure 5, including the step-wise [68], cosine [8], and our exp-decay learning rate schedule. Basically, our schedule is an early-stop version of the baseline of TRADES [68], which skips the small learning rate stage as soon as possible in the later stage. We found this schedule is the most effective one when only training on the original CIFAR10. However, when combined with the 500K unlabeled images from RST [8], we find that the over-fitting problem is much less severe and cosine learning rate is the best choice.

## D    Boosting the Original Adversarial Training

We further show that our strategy also applies to the original adversarial training [39]. Note that our generalized adversarial training framework (1.1) allow us to further boost the robust regularization for original (generalized) adversarial training. The only caveat is that in adversarial training formulation, the robust regularization term is not guaranteed to be non-negative in practice[3]. To avoid this problem, we manually set the robust regularization term in (1.1) to be non-negative by clipping the $\mathcal{L}(\boldsymbol{\theta}; \widehat{\mathbf{x}}, y) - \mathcal{L}(\boldsymbol{\theta}; \mathbf{x}, y)$ term. Let us denote $\mathbf{x}'$ as the empirical maximization solution, the final loss function becomes:

$$\underset{\boldsymbol{\theta}}{\arg\min} \, \mathbb{E}_{(\mathbf{x}, y) \sim \mathcal{D}} \Big\{ \mathcal{L}(\boldsymbol{\theta}; \mathbf{x}, y) + \lambda \cdot \max_{\widehat{\mathbf{x}}_i \in \mathbb{B}(\mathbf{x}_i, \epsilon)} \big( \mathcal{L}(\boldsymbol{\theta}; \mathbf{x}', y) - \mathcal{L}(\boldsymbol{\theta}; \mathbf{x}, y), 0 \big) \Big\}.$$

The bottom part of Table 1 shows the experimental results for boosting the robust regularization parameter for (generalized) adversarial training models. We can observe that the boosting strategy still works in this method, and wider models indeed require larger $\lambda$ to obtain the best robust accuracy.

## E    Verifying Our Findings on ImageNet

We further test the model of Fast AT [63] on ImageNet dataset in Table 5, and it again verifies our conclusion that larger model width would increase natural accuracy but decrease perturbation stability.

Table 5: Fast Adversarial Training on ImageNet.

| Models | $\lambda$ | Robust Accuracy | Top5-Natural Accuracy | Perturbation Stability |
|---|---|---|---|---|
| WideResNet-50-1 | 1.0 | 38.34 | 53.24 | 72.29 |
| WideResNet-50-2 | 1.0 | 51.65 | 66.67 | 70.10 |

---

[3]Successfully solving the inner maximization problem in (1.1) is supposed to guarantee that $\mathcal{L}(\boldsymbol{\theta}; \mathbf{x}', y) > \mathcal{L}(\boldsymbol{\theta}; \mathbf{x}, y)$, however, in practice, there still exist a very little chance that $\mathcal{L}(\boldsymbol{\theta}; \mathbf{x}', y) < \mathcal{L}(\boldsymbol{\theta}; \mathbf{x}, y)$ due to failure in solving the inner maximization problem at the beginning of the training procedure with limited steps.

## F  Boosting the Regularization Parameter on Extra Adversarial Training Methods

We also compare with other models from the AutoAttack [15] leaderboard. We focus on the AWP [64] and show the result in Table 6. We found that our conclusion still holds for the AWP method that using larger $\lambda$ (12.0 rather than 6.0 in the default setting) can achieve even better robust accuracy.

Table 6: AWP on CIFAR10 dataset.

| Models | $\lambda$ | Robust Accuracy | Natural Accuracy | Perturbation Stability |
|---|---|---|---|---|
| WideResNet-34-10 | 6.0 | 59.01 | 84.82 | 73.95 |
| WideResNet-34-10 | 12.0 | 59.34 | 81.20 | 76.65 |
| WideResNet-34-10 | 18.0 | 58.72 | 78.43 | 77.54 |

## G  Evaluating the Three Metrics on State-of-the-Art Models

In the figure below, we evaluate nine state-of-the-art robust models against the PGD attack for the three metrics: the natural accuracy, the perturbation stability, and the robust accuracy (the size of the ball). Our dissection of these three metrics helps the researcher better understand how different approaches influence adversarial robustness. For instance, we can tell that HE [44] mainly helps the stability, Pretrain [28] mainly helps the natural accuracy and slightly hurts stability. Moreover, we can tell that methods like RST[8] simultaneously improve the natural accuracy and perturbation stability. This observation shows that it is possible to improve the two contradictory metrics.

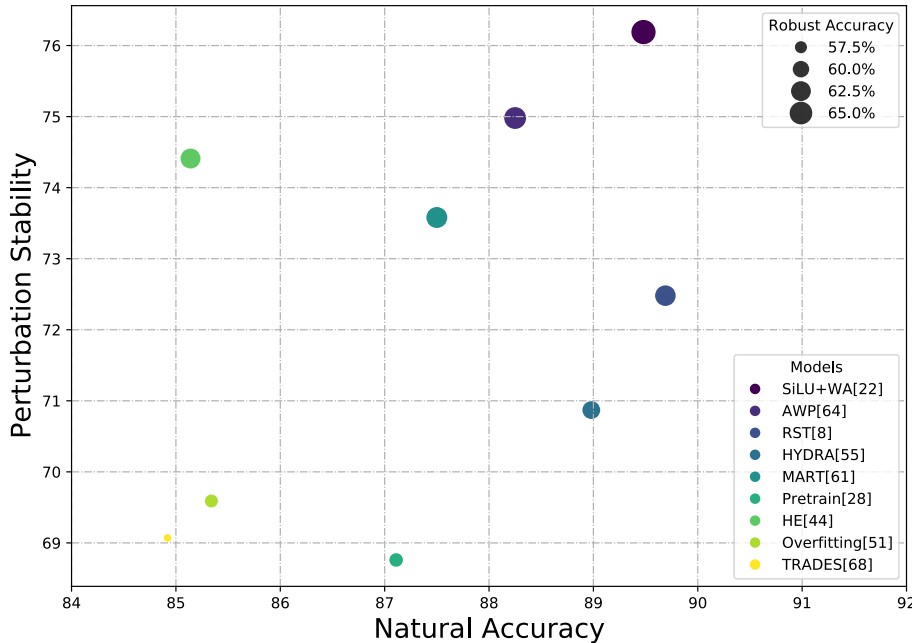

Figure 6: The changing trend leanring rate against training epochs for different learning rate schedule.

# H More Illustrations of Eqn. (1.1)

In this part, we provide a complete visualization for the two parts in Eqn. (1.1). The figures below are an extension of Figure 1, where the models are those we trained in Table 2. We test WideResNet-34 on CIFAR10 and CIFAR10. We test DenseNet-BC-40 on CIFAR10. The two losses with respect to different robust regularization parameter $\lambda$ are shown. Again, we emphasize that the observation that wider neural networks achieve worse performance on stability with the same $\lambda$ can be found during the training stage. Therefore, this intriguing phenomenon is not an over-fitting problem, as previous works [51] pointed out.

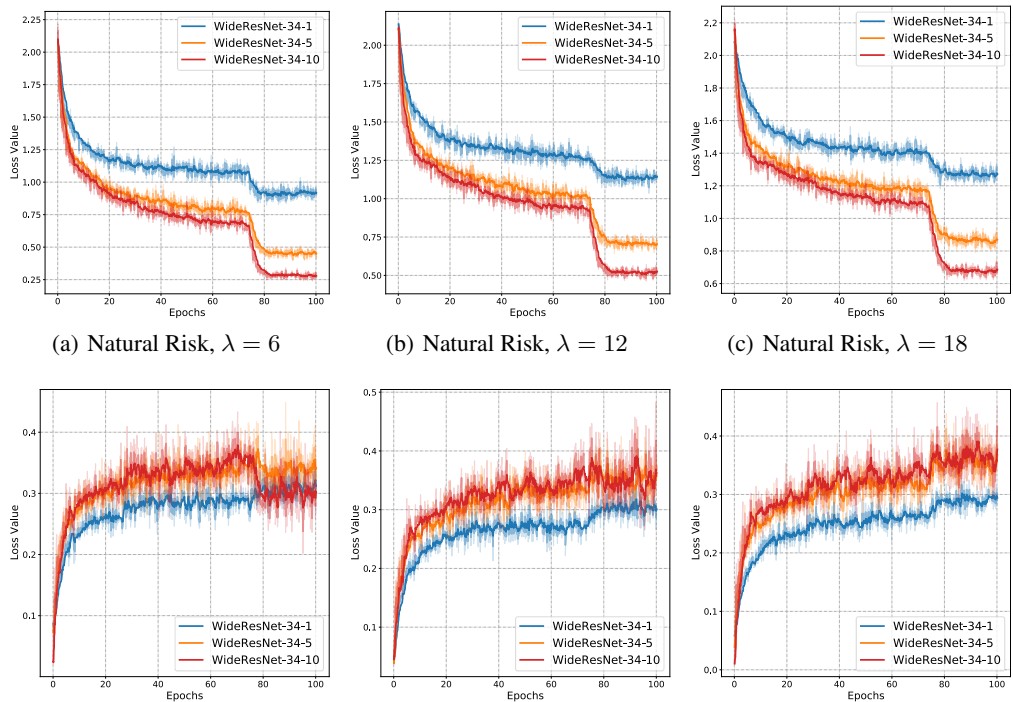

(a) Natural Risk, $\lambda = 6$   (b) Natural Risk, $\lambda = 12$   (c) Natural Risk, $\lambda = 18$

(d) Robust Regularization, $\lambda = 6$  (e) Robust Regularization, $\lambda = 12$  (f) Robust Regularization, $\lambda = 18$

Figure 7: WideResNet-34 on CIFAR10.

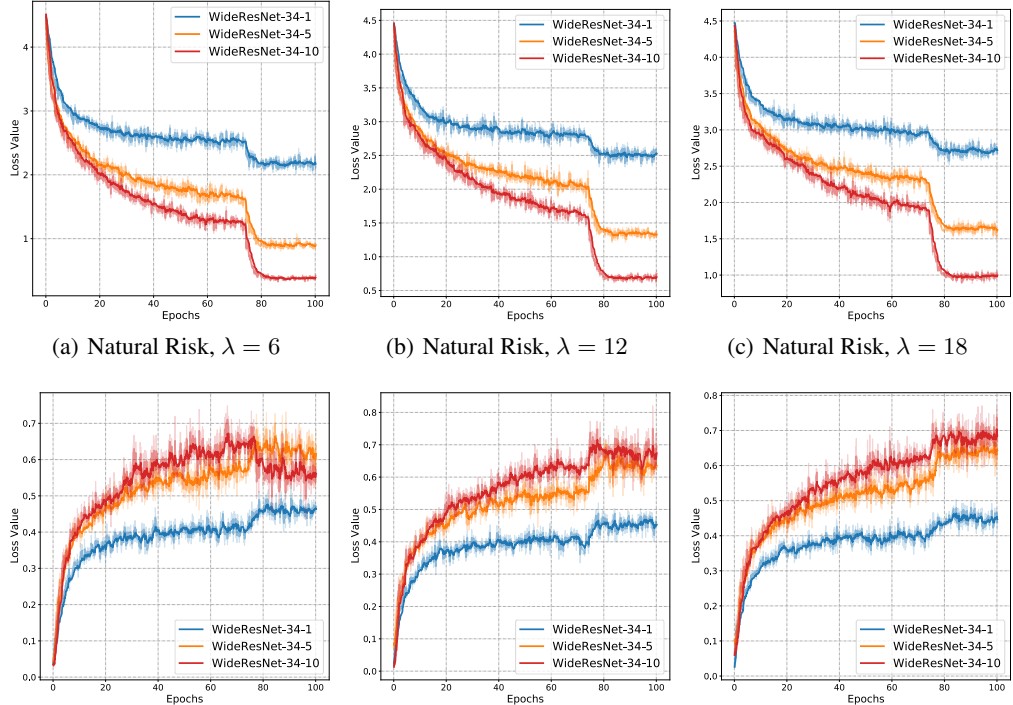

(a) Natural Risk, $\lambda = 6$ (b) Natural Risk, $\lambda = 12$ (c) Natural Risk, $\lambda = 18$

(d) Robust Regularization, $\lambda = 6$ (e) Robust Regularization, $\lambda = 12$ (f) Robust Regularization, $\lambda = 18$

Figure 8: WideResNet-34 on CIFAR100.

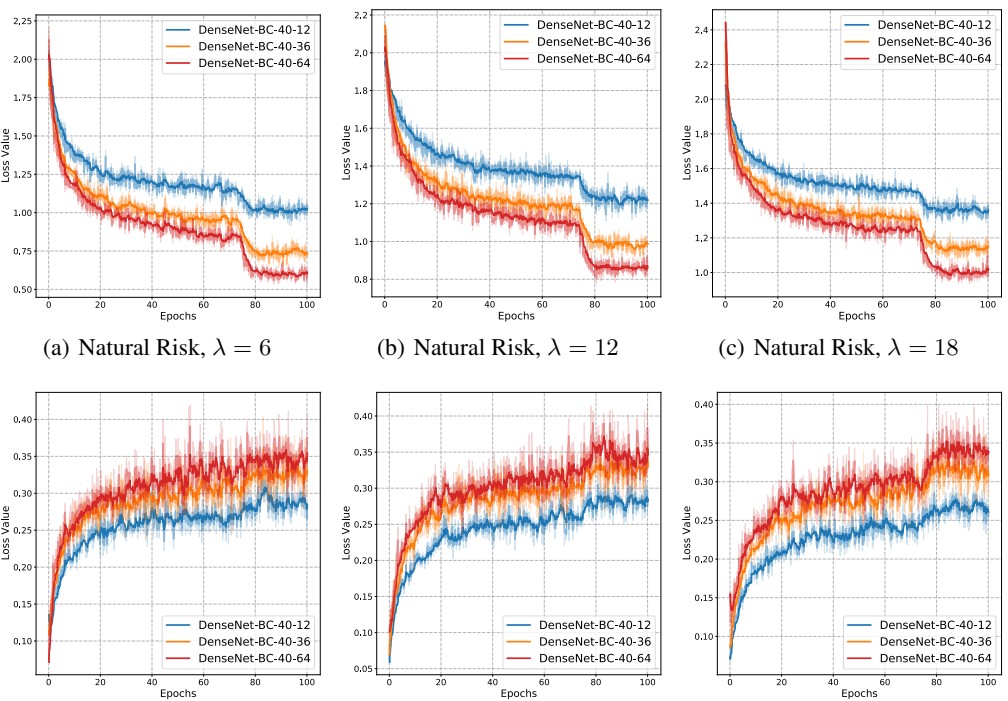

(a) Natural Risk, $\lambda = 6$ (b) Natural Risk, $\lambda = 12$ (c) Natural Risk, $\lambda = 18$

(d) Robust Regularization, $\lambda = 6$ (e) Robust Regularization, $\lambda = 12$ (f) Robust Regularization, $\lambda = 18$

Figure 9: DenseNet-BC-40 on CIFAR10.