# OpenReview forum: "Do Wider Neural Networks Really Help Adversarial Robustness?"
_NeurIPS.cc/2021/Conference — NeurIPS 2021 Poster_

### Official Review · Reviewer_ej3Q · 2021-07-05

**Rating:** 7
**Confidence:** 3

**Summary:**

In this paper, the authors evaluate the impact of network width on the robustness of models trained through adversarial training. The authors (1) formulates robust accuracy as the intersection of natural accuracy and perturbation stability and show that there is tradeoff between them (2) theoretically show that increasing network width leads to higher local Lipschitz constant which leads to lower perturbation stability, (3) empirically show that networks with larger width need a larger robust regularization hyperparameter to perform well and propose a method (WAR) to tune this value.



**Limitations And Societal Impact:**

I do not see any discussion of limitations.  A potential concern is how well WAR performs on other architectures/datasets.

**Main Review:**

Originality: To the best of my knowledge, the contributions of this submission are original. Prior works have indicated that adversarial training performs better with models of larger capacity, but the authors explore the impact of increasing the model width and empirically and theoretically show that this is not necessarily the case.  The authors do a good job of citing related works.

Quality: The quality of the submission is good.  The authors provide theory as well as rigorous experiments to show that higher model width leads to lower perturbation stability and the need for a higher robust regularization value.  The scope of these experiments is good, the authors experiment with multiple architectures (WideResNet, DenseNet-BC) as well as different adversarial training techniques (TRADES, Adversarial Training), datasets (CIFAR-10, CIFAR-100, Imagenet), and different attack methods.  For the section where the authors introduce WAR, I can only see experiments on WideResNet and what I think is the CIFAR-10 dataset.  I think the authors can benefit by also extending these experiments to different architectures and datasets as well.

Clarity: Overall the paper is clear, but the paper can be proofread for grammar.  Some comments:
- try to use in text citations (\citet{}) when using the citation as a subject or object in a sentence
- I think the first paragraph under section 3.1 can be placed with related works
- line 163: "unlearned model" --> "untrained model"
- last sentence of 1: "... such that the learned models cannot be easily perturbed for the sake of model robustness" --> this wording is awkward.  I think what is meant is that the models aren't fooled by small perturbations which in turn improves model robustness
- 171-173: "Works including [49] and [43] also challenged this robust-natural trade-off [59] does not hold for some cases" --> awkward wording and "does not hold for some cases" is very vague and can be elaborated on
- 173-174: "Therefore, we argue ... of this trade-off"-->  these 2 sentences should be 1 sentence joined by a comma
- 179-180: "Do they only help natural risk, to robust regularization, or maybe both of them." --> should be combined with the previous sentence and should be a question
- line 182: "explains why the" --> "illustrates how"
- line 183: "it still remains unclear what is the underlying reasons ..." --> "it still remains unclear what the underlying reasons are ..."

Significance: The results lead to a better understanding of how width impacts the robustness of neural networks.  The authors also propose WAR to finetune the robust regularization hyperparameter for models of larger width which maintains the same tuning time that is used in practice while improving robustness for these larger models.

**Time Spent Reviewing:**

2

---

> ### Author Response · Authors · 2021-08-10
> **To Reviewer ej3Q**
>
> Thanks for your thorough review and your positive feedback for our work!
>
> ---
>
> ### Point 1 (WAR on other architectures and datasets)
>
> Thanks for your suggestions. We will add experiments of WAR on DenseNet-BC and CIFAR100 in the camera-ready. Here we present some preliminary results:
>
>
>
> Apply WAR to DenseNet-BC on CIFAR10.
>
> | - | DenseNet-BC-40-12 / $\lambda$ | DenseNet-BC-40-64 / $\lambda$ |
> | ---- | ---- | ---- |
> | Manual Tuning | 44.71% / 6 | 55.79% / 12 |
> | WAR | 44.70% / 5.84 | 55.86% / 11.32 |
>
>
>
> Apply WAR to WideResNet on CIFAR100.
>
> | - | WideResNet-34-1 / $\lambda$ | WideResNet-34-5 / $\lambda$ | WideResNet-34-10 / $\lambda$ |
> | ---- | ---- | ---- | ---- |
> | Manual Tuning | 24.27% / 6 | 31.57% / 12 | 32.88% / 18 |
> | WAR | 24.28% / 6.17 | 31.55% / 13.20 | 32.90% / 17.79 |
>
>
>
> We can observe that WAR performs consistently well across different architectures and datasets.
>
>  ---
>
> ### Point 2 (grammar issues)
>
> Thanks a lot for your efforts to help us improve our work! We will refine all these points based on your suggestions.

---

> > ### Comment · Reviewer_ej3Q · 2021-08-14
> > **WAR generalizability**
> >
> > Thank you for providing additional experiments for WAR on other architectures/datasets to show that it generalizes well.  Since this was my main concern, I will raise my score to 7.

---

> > > ### Author Response · Authors · 2021-08-15
> > > **Thank you**
> > >
> > > Thank you for your positive feedback and for increasing the score. We will be sure to include the additional experiments in the final version of the paper.

---

### Official Review · Reviewer_vSzi · 2021-07-16

**Rating:** 7
**Confidence:** 3

**Summary:**

The paper discusses how the width of ResNets related to their robustness. They do adversarial training with a hyper-parameter for the strength of the robust regularization and show that making the model wider while keeping the hyper-parameter constant will increase clean accuracy but reduce the robust generalization.

They define a new metric, perturbation stability, which is the rate of samples keeping their label prediction under adversarial attacks, even if the label prediction is wrong without pertubations. The authors connect this metric to the local Lipschitzness and use it to study how the width of a network affects it. They show that both through an analytical calculation and empirical results that the local Lipschitz constant increases with the width of a network.

In further experiments the authors show that wider networks need a higher weight on the robust regularization loss to get optimal robust accuracy, while previous work often optimized this weight for smaller networks and then used the same value after increasing the width. The results are verified for other datasets and architectures and finally the authors present a strategy to adaptively choose the hyper-parameter for wider models by first optimizing it on smaller models and then optimizing during training to achieve the same ratio between natural loss and robust regularization loss for the wider model that was optimal for the smaller model.


**Limitations And Societal Impact:**

The authors have addressed limitations and societal impact adequately.

**Main Review:**

I think the paper is well written and gives a new perspective on adversarial training of wide networks. The authors do a good job of exploring the topic from different directions:
- Given a motivation why they suspect better tuned $\lambda$ on wider models would help improve robustness
- Giving analytical insight through the Lipschitz constants.
- Showing the improvement in robust accuracy through empirical results.
- Proposing a method to automatically tune $\lambda$ for wider models while training.

Because of this I think this paper should be accepted to Neurips.

Some things that could be improved:
- In line 64 and at later points the authors say that increasing the width of models while keeping $\lambda$ constant would decrease the overall robustness. I find the a bit inaccurate as robustness is usually looked at in terms of the robust accuracy and it (as the authors show in figure 3) usually increase with the width. I think the authors should clarify that they mean the perturbation stability.
- The authors in their analysis in section 4 show that $\lambda$ should grow with $\sqrt{mH}$ but don't use that later on. Could this be an easier way to optimize $\lambda$?

Minor comments:
- Line 3 "it remains elusive how does neural network width affect model" should be "it remains elusive how neural network width affects model"
- Line 45 same as above
- The notation in line 81-82 seems unnecessary. The defined notation is very standard and other similarly standard notation isn't explained.
- Line 84 "tremendous work" instead of "tremendous works"
- Line 216 "increases with" instead of "increases as"
- Line 264 "90-th epoch as in" instead of "90-th epoch in"

**Time Spent Reviewing:**

5

---

> ### Author Response · Authors · 2021-08-10
> **To Reviewer vSzi**
>
>
> Thanks for your thorough review and positive feedback for our work!
>
>  ---
>
> ### Point 1 (line 64)
>
> We apologize for the confusion here. In Line 64, we first show that "keeping $\lambda$ constant often worsen **stability**". This is verified by Figure 3(c). Then we further stated that "keeping $\lambda$ constant lead to **possible** decrease of robust accuracy/robustness". As shown by Figure 3(a) and the top part of Figure 3(d), for robust accuracy, the width-5 model slightly outperforms the width-10 model when using constant $\lambda$. Therefore, keeping $\lambda$ constant can indeed lead to the decrease of robust accuracy (may not always lead to but can lead to). We hope this solves your concern and we will try to make it more clear in the camera-ready.
>
>  ---
>
> ### Point 2 (lambda and sqrt(mH))
>
> It would be nice if the theoretical result can directly help us pick the best network width. However, we want to emphasize that the $\sqrt(mH)$ bound is for the local Lipschitz constant but not for $\lambda$. In summary, the theory tells us that $\lambda$ should grow but we do not know what rate it should grow (as $\lambda$ is for balancing the surrogate losses while the local Lipschitz constant directly affects the accuracies).
>
>
> ---
>
> ### Point 3 (minor comments)
>
> Thanks for pointing out these issues! We will refine all these points based on your suggestions.

---

> > ### Comment · Reviewer_vSzi · 2021-08-16
> > **Thanks**
> >
> > Thanks for the rebuttal, especially the explanation why the theoretical results can't be used to set lambda directly.

---

> > > ### Author Response · Authors · 2021-08-25
> > > **Thank you for your support!**
> > >
> > > We're glad that our explanation resolves your question. We will revise our paper according to your and other reviewers' comments.

---

### Official Review · Reviewer_7XmE · 2021-07-17

**Rating:** 7
**Confidence:** 5

**Summary:**

The paper studies the connection between adversarial robustness and model capacity in the form of width. The key idea is to decompose robust accuracy into standard accuracy and stability---i.e., on what fraction of test inputs the prediction of the model cannot be changed by an adversary.

The authors find that while increased width tends to improve robust accuracy (as has also been previously observed) the gain are mostly coming from an improvement in the standard (test) accuracy of the model, while the stability of wider models is actually _decreased._ To counteract this phenomenon, they suggest increasing the regularization term corresponding to stability as the width of the model increases and indeed observe empirical performance improvements. The authors further support their hypothesis by theoretically analyzing the effect of width on robustness in an NTK setting and observing similar results. Finally, they propose a heuristic scheme for adjusting the stability regularization parameter during training and demonstrate promising results.

**Limitations And Societal Impact:**

Yes, they do.

**Main Review:**

Overall, this paper makes a significant contribution on an important topic---we still do not fully understand the connection between capacity and robustness. The perspective proposed is simple and intuitive but at the same time appears to be quite useful for reasoning about the empirical behavior of wide models. I believe that these results will be of interest to anyone who cares about robustness.

Questions and comments to authors (not affecting score):
- Figure 1 and 3: What is epsilon/norm used for training and eval?
- Did you perform additional experiments with different norms and values of epsilon? Were the results consistent?
- L167 (and similar discussion in other places): In all three works mentioned [68,59,50], the underlying trade-off actually manifests as a stability-accuracy trade-off. Specifically, in [68, 59] the trade-off manifests because robust models cannot rely on predictive but unstable features, while in [50] the trade-off is a precisely a result of the requirement that the model's prediction needs to be stable around training points. Thus, while this discussion appears to be contradicting these prior works, it actually doesn't.

**Time Spent Reviewing:**

3

---

> ### Author Response · Authors · 2021-08-10
> **To Reviewer 7XmE**
>
> Thanks for your thorough review and positive feedback for our work!
>
>  ---
>
> ### Question 1 (epsilon/norm)
>
> We adopt the same settings as TRADES in Figure 1 and Figure 3. Namely, we set the epsilon to be 8/255 under the $L_\infty$ norm for both training and testing.
>
>  ---
>
> ### Question 2 (different norm and epsilon)
>
> Thanks for your suggestions. We will add such experiments in the camera-ready. Here we present some preliminary results.
>
> We verify our findings with respect to the L2 norm. In detail, we set the $\epsilon$ to be 0.5 and $\lambda$ to be 6. The results below are consistent with our finding for the $L_\infty$ norm.
>
> | Metric | WideResNet-34-1 | WideResNet-34-10 |
> | ---- | ---- | ---- |
> | Natural Accuracy | 81.52% | 89.62% |
> | Robust Accuracy | 65.11% | 70.04% |
> | Perturbation Stability | 83.01% | 81.22% |
>
>
>
> Due to the tight time constraints, we are unable to retrain all the experiments with different $\epsilon$ before the rebuttal period ends. We will continue working on it and bring more comprehensive results in the camera-ready.
>
>
>
>  ---
>
> ### Question 3 (L167)
>
> We agree with the reviewer that these previously works’ conclusions do not contradict ours from the inside. In the paper, we just want to emphasize that previously the concept of perturbation stability and robustness are usually confused and not well-separated. We thank the reviewer for the comments, and we will refine this discussion based on your suggestions.

---

> > ### Comment · Reviewer_7XmE · 2021-08-25
> > **Thank you for the response**
> >
> > I appreciate the author's response as well as the preliminary results provided.
> >
> > I still believe that this is a significant paper and I thus leave my score as is.

---

> > > ### Author Response · Authors · 2021-08-25
> > > **Thank you for your support!**
> > >
> > > We will be sure to revise our paper accordingly in the final version.

---

### Official Review · Reviewer_TGwT · 2021-07-17

**Rating:** 5
**Confidence:** 4

**Summary:**

The authors investigate the relation between network width and adversarial robustness. There are some attempts to connect to some theory, but the results are largely empirical.


**Limitations And Societal Impact:**

Yes

**Main Review:**

- The main problem I find with the current paper is that the main message is not clear. There are some empirical comparisons based on the width of neural networks in relation to adversarial robustness, and some theorems based on previous results on Lipschitz bounds on neural networks. But the main novelty or contribution of the paper is not clear.

- The only main message I could find is that increasing the width does not necessarily add to the adversarial robustness, contrary to the claims of some of the previous works. But this alone is not a very significant contribution.

- There is a problem with analyzing the relation between natural accuracy and robust accuracy using equation 3.1. Equation 3.1 is based on accuracy, whereas in practice we always optimize a surrogate loss function such as the cross-entropy loss. The tradeoff in the optimization is not the direct tradeoff of accuracies, but tradeoff of their respective loss functions. Given the cross-entropy loss can be unbounded, the actual tradeoff might not be similar to equation 3.1.

- The analysis of perturbation stability using Lipschitz constant in section 4 is not particularly new. And apart from stating the bounds, there is little insight on how the bounds affect the tradeoffs and relate to the empirical results.

- The rest of the paper is on empirical comparison of width vs adversarial robustness. But without better connections to other parts of the paper they seem like standalone empirical results.

- I believe the authors need to improve on the clarity of presentation of their paper, and state more clearly what their actual contributions are.


**Time Spent Reviewing:**

3

---

> ### Author Response · Authors · 2021-08-10
> **To Reviewer TGwT**
>
> Thanks for your valuable opinions. We will address all your concerns in the following order:
>
>   ---
>
> ### Clarity of main message (Points 1/2/6)
>
> The main message for our paper is two-fold, as we illustrated in the abstract. First, as you also mentioned, increasing the width does not necessarily add to the adversarial robustness. Second, one needs to properly enlarge $\lambda$ to unleash the robustness potential of wider models fully. This is built on the first message to provide some actual guidance for practitioners. And we also provide a practical algorithm (WAR) to achieve this. Combining all these, we believe our work here would certainly help practitioners who want to train large-scale robust models with much less wasted time in tuning parameters.
>
>   ---
>
> ### About Eqn 3.1 (Point 3)
>
> We do agree that the two trade-offs cannot be exactly the same (but they are closely related). In fact, we have empirically verified the trade-off for both training loss (Eqn 1.1 and Figure 1) and testing accuracy (Eqn 3.1 and Figure 3) and they showed similar patterns.
>
> However, we respectively disagree with your word “actual trade-off", which we believe you refer to as the surrogate loss trade-off. Yet we believe that the accuracy trade-off is more fundamental and should be our focus here, as TRADES [68] and many other works also did.
>
>  ---
>
> ### The analysis of Lipschitzness (Point 4)
>
> > using Lipschitz constant is not particularly new and there is little insight on how the bounds affect the tradeoffs
>
> We believe there are some misunderstandings here. We did not claim to build a new proof technique to analyze the Lipschitz constant, instead, our goal here is to figure out the relationship between network width and model robustness, while this Lipschitz bound serves as an important link for this relationship. As we mentioned in Section 4.1, a smaller local Lipschitz constant would lead to better perturbation stability. This is exactly how the Lipschitz bound affects the trade-off.
>
>   ---
>
> ### Empirical results (Point 5)
>
> > without better connections to other parts of the paper they seem like standalone empirical results.
>
> We are sorry that we did not explain our experimental logic clearly enough for you. Our empirical study still closely follows our main messages. In Section 4, we showed that wider networks lead to worse perturbation stability. So, the first part of the experiments focuses on verifying whether this claim is true in practice and would increasing $\lambda$ help the situation here. Given these theoretical and empirical studies, we can conclude that one needs to properly enlarge $\lambda$ to fully unleash the robustness potential of wider models. And then in the second part of our experiments, we provide a practical algorithm (WAR) to achieve this goal automatically and verify its effectiveness.

---

### Decision · Program_Chairs · 2021-09-27

**Decision:**

Accept (Poster)

**Comment:**

The reviewers had raised a few concerns which were mostly resolved after the authors' response. All the reviewer agreed in the discussions that the contributions of the paper are important and interesting. Also, the reviews contain all the points mentioned in the discussions, so there is nothing more to add here.